# Assembling Cultural and Natural Values in Vernacular Landscapes: An Experimental Analysis

**Pablo Altaba** [1] , **Juan A. García-Esparza** [1,*] **and Anna Valentín** [2]

1   School of Technology and Experimental Sciences, Department of Mechanical and Engineering Construction, Universitat Jaume I, 12071 Castelló de la Plana, Castelló, Spain
2   SEO/BirdLife, Territorial Office in Comunidad Valenciana, C/Tavernes Blanques 29, Bajo, 46120 Alboraia, Valencia, Spain
*   Correspondence: juan.garcia@uji.es

**Abstract:** Cultural landscapes can host natural and cultural areas. However, often, this distinction is not clear cut and the attempts to clarify this blur the character of landscapes. Vernacular landscapes today act as a living legacy, subject to transformation, preservation, or abandonment. This study analyses these legacies in order to evaluate elements and interactions. The research uses GIS with spatial and thematic databases of cultural heritage and natural habitats, as well as open data, historical cartography, citizen participation, and fieldwork information sources. In combination with GIS tools, LiDAR images helped in the dataset evaluation process. A priority scale of conservation for different areas was outlined through a process cataloguing the natural and cultural assets with conservation and intervention rubrics. These settings are classified according to their cultural and natural value, conservation, surrounding environment, and potential threats. The experimental methodology of this study aims to add new options for characterising vernacular landscapes by adding soft participatory values to datasets. These prove to be reliable complementary information, improving accuracy.

**Keywords:** cultural; natural heritage; GIS; spatial databases; vernacular landscape; heritage database; landscape conservation

## 1. Introduction

Cultural landscapes are areas of land which have been shaped by traditional human land use. They host historical and contemporary natural and cultural values [1,2]. In both cultural and natural terms, vernacular architecture tells the tale of the landscape history of interactions [3], the result of tangible and intangible forms of human behaviour in rural environments [4,5]. The extensive research and studies carried out in recent decades [6,7] show how the loss of natural and cultural biodiversity has become a topic of increasing concern [8].

One of the main tools for recording this biodiversity is cataloguing. In the cultural realm, Ruggiero et al. [9] and Fuentes [6] see cataloguing processes as an analysis of the potential to assess the reuse of cultural assets. In parallel, while many authors refer to other processes, such as landscape characterisation [10–12], Brown et al. [13] saw both cataloguing and characterisation as the acquisition of the collective understanding of a place. Different academic disciplines, including architecture, archaeology, ecology, geography, and history, employ distinctive approaches to the study of changes observed in historic landscapes [14]. Consequently, the effectiveness of methodologies relies on adaptation to and co-existence with complementary disciplines.

Today, almost all fields of inquiry approach the physical space through Geographic Information Systems (GIS). This helps create a common ground of inquiry for the characterisation of places and decision-making. It is particularly useful for the analysis of potential conflicts of interest between natural and cultural interactions [9,15,16]. In the case currently under examination, it helps debate on forms of conservation [8,17] and management [18,19].

In the end, this tool, as Statuto et al. [20] explained, may help balance landscapes with multiple natural and anthropogenic factors interacting.

When referring to landscape characterisation, it is not easy to find approaches that combine a natural and cultural approach with a management purpose [21,22]. As different administrations have independent catalogues, it remains a challenge to obtain combined datasets of geo-located information to enable more accurate and interrelated analysis. The lack of joint approaches makes analysis partial and, therefore, unbalanced [23–26]. However, sources are quite varied since landscape assessment combines the use of old maps [27], open digital datasets [28], and that of media applications through GIS [29–31]. Linked to this, as Brown and Weber [32] point out, public participation in mapping for value elicitation is an essential part of the process. Nonetheless, although local spatial knowledge remains an essential source of information for investigating the landscape, when locals explain their own perspective, they are unaware of or oblivious to the relative relevance of their vision when examined within the full spectrum of elements and values [33,34].

Thus, the critical challenge faced in this study is the joint evaluation of cultural and natural assets to inform a conservation plan that is agreed upon by all participants and not just informed. The added value of this research lies in its use of cataloguing based on the geopositioning of building elements, which it combines with biodiversity-related factors, such as habitats, protections, and landscape zoning. Accordingly, the objective of this research is two-fold. On the one hand, researchers assess a methodology, which allows them to record cultural and natural assets and document and combine conservation issues. The researchers analyse the potential of ensembles linked to places of special interest, in which both the conservation of cultural and natural elements transversally mirror and pivot on the preservation of other intangible assets. Equally, the use of a GIS tool helps researchers to create semi-automatic procedures. Ultimately, the overall aim is to inform work with the conservation guidelines stemming from current policy. Based on these objectives, the article provides a detailed explanation of the approach to the area of study, the process to obtain data, the methodology (both on-site and desk-based), presents the results, and sets out the starting points and limitations for a conservation plan.

## 2. Area of Study

This study focuses on the northern mountain areas of the Valencia Region, located in the east of Spain (Figure 1). The orography varies from 0 to 1814 m.a.s.l. The area under study, extracted from a study of visual basins of the orography, has a surface of 269.16 km². This territory is known locally as the Penyagolosa Mountain area. The combined population of the ten villages that form this landscape does not exceed 5000 inhabitants. The area hosts remains of cultural and natural consolidated legacies, dating back to the earlier periods when Penyagolosa was first occupied during the 13th century. Since then, the cultural landscape has stratified and the natural landscape has evolved accordingly. Nowadays, this landscape is experiencing a gradual but steady process of depopulation.

The area is home to a number of cultural elements, most notably dry-stone infrastructure. This study examines different communication itineraries, paths, or cattle trails used to move across the territory. These itineraries are delimited by dry-stone walls, stone bridges for crossing rivers, and several patches of irregular paving on the more abrupt slopes. Other constructions, such as fountains, troughs, wells, and ponds, were used to collect water and to provide drink for livestock. There are also other constructions linked to the use of water: flour mills, small industries equipped with water-powered machinery, and auxiliary waterwheels for the mass extraction of water from the subsoil using animal traction.

Furthermore, there are farmhouses with terraced plots scattered throughout the area and associated to previous forms of farmland and livestock. Some of them stood out as they performed additional functions as mills, workshops, or even fortifications, with small towers or palisades to help the population in the event of conflict. Other characteristic constructions found in the area are religious: hermitages used by farmers for worship.

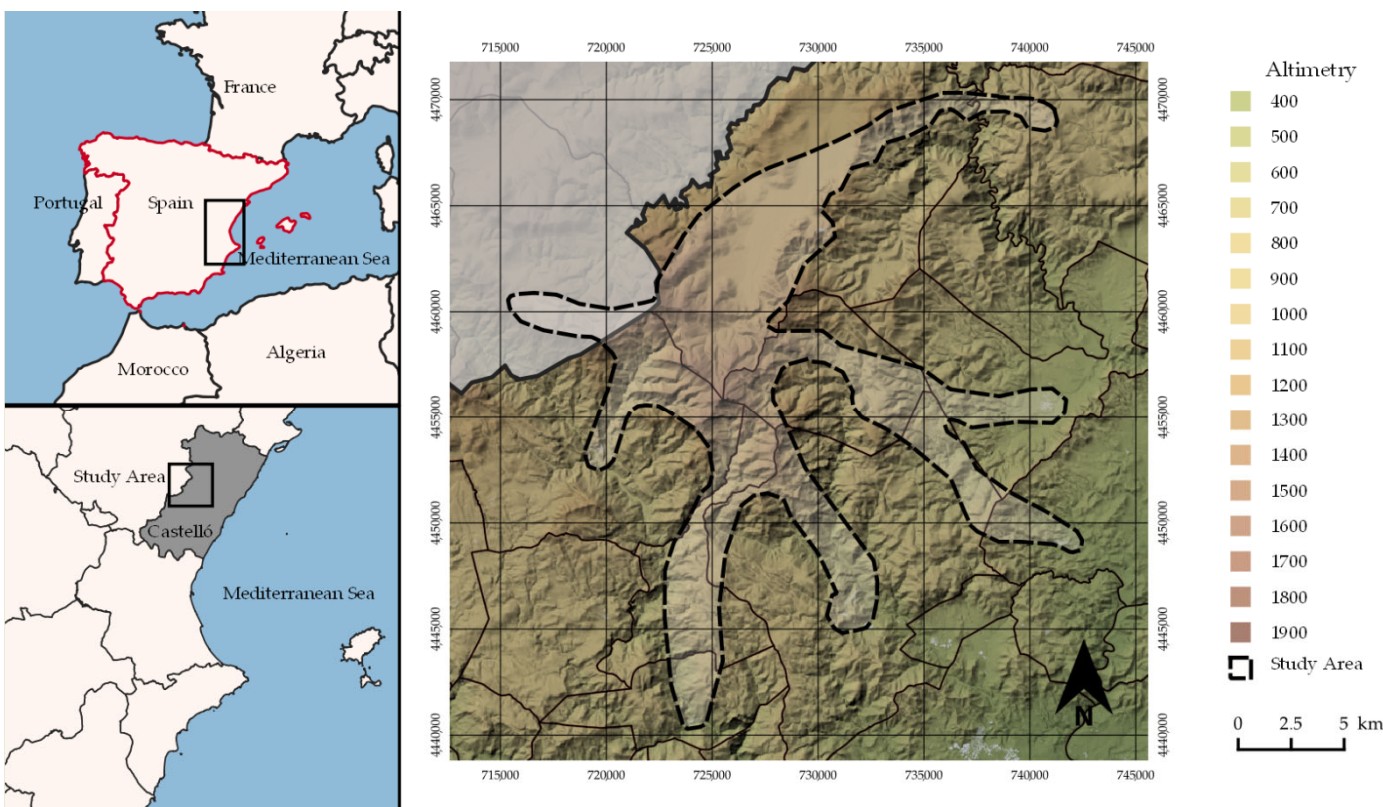

**Figure 1.** Location and delimitation of the study area and orographic characteristics of the province of Castelló. Source: authors.

From a geological perspective, the Penyagolosa Mountain range features elements from the Cretaceous, Jurassic, and Triassic periods of the Mesozoic era. Geologically, it dates from 66 to 250 million years ago. The four bioclimatic floors of the Mediterranean as set out by the altitudinal scale (the Thermo-Mediterranean, the Meso-Mediterranean, the Supra-Mediterranean, and the Oro-Mediterranean) are present in the study area.

The biota in this area are dominated by vegetation formations of holm oak (*Quercus ilex subsp rotundifolia*) and Aleppo pine (*Pinis halepensis*) below an elevation of 1000 metres, while pine groves of black pine (*Pinus nigra*) and maritime pine (*Pinus pinaster*), as well as juniper (*Juniperus thurifera*), proliferate above. The presence of Valencian oak (*Quercus faginea*), yew (*Taxus bacata*), and lime (*Tilia platyphyllos*) should also be highlighted, as well as the interesting formations of Pyrenean oak (*Quercus pyrenaica*) in siliceous soil.

The Natural Resources Management Plan of the Penyagolosa Natural Park specifies that there are certain species of fauna considered to be of particular interest. This includes invertebrates, such as the marsh fritillary (*Euphydryas aurinia*) [35], and 24 species of reptiles and amphibians, 15 of which are linked to agricultural environments. It is also possible to find mammals, such as the mountain goat, Pyrenean goat, different chiropteran species, and 131 species of birds, 46 of which are linked to agricultural mosaics [36].

Apart from the Natural Park, the study area has a high ecological value, is part of the Nature 2000 Network, and hosts several habitats of community interest, some of which are considered priorities in terms of conservation. In this sense, it has been declared a Special Protection Area for Birds (SPA) and Site of Community Importance (SCI). This territory is also home to constructions made using the dry-stone technique and declared UNESCO Intangible Assets of Local Importance and World Heritage. These protections stem from contemporary legislation at three different levels: international, national, and regional, and involve three different contexts: natural heritage, cultural heritage, and landscape heritage.

The review of legislative documents reflects no regulatory mechanisms, which combine the interests of cultural and natural heritage. The regional regulations feature figures

that aim to unite some aspects of these two fields. However, at the moment of writing this paper, there is no legislative evidence to serve as a guide for the analysis of this case.

## 3. Materials and Methods

In this context, in order to elaborate a methodology that links architectural–ethnographic elements with natural values, it is essential to know the sources of information available. In this case, the study was drawn up from five sources: historical maps and archives, citizen participation, fieldwork, and open databases. Based on this, the scientific value of raw data collected was further analysed to make these scalable to other locations. The main objective of this phase was to determine the communication itineraries along which natural and cultural heritage merged in a shared space. To do so, the work was divided into three parts: documentation, fieldwork data collection, and GIS analysis (Figure 2).

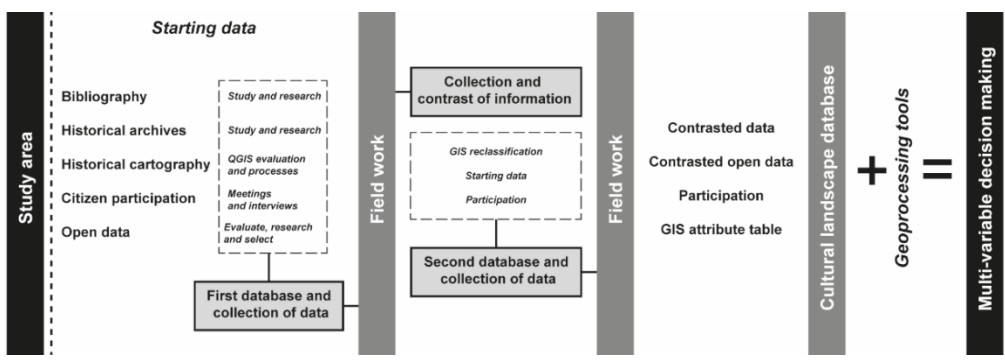

**Figure 2.** Flowchart of the cultural assessment procedure applied to heritage sites. All processes are described in detail in Sections 3.1 and 3.2.

### 3.1. Starting Data

#### 3.1.1. Historical Mapping

The search for historical mapping began with a limitation. Information on depopulated rural regions in Spain and in the region of Valencia shows weak territorial structures. This is evident due to the lack of historical evidence on cartographic resources on a micro-territorial scale (1:10,000 to 1:50,000) until well into the 20th century. The only sources found from the first decades of the 20th century are available in an open format, compiled and geo-referenced by the National Centre of Geographic Information (CNIG) in both image and referenced raster ECW format. As these maps used the ED50 reference system researchers adapted them to the European ETRS89 system.

These maps are topographical works by the CNIG with an approximate scale of 1:25,000 and although they lack a structured interpretation, due to their precision it is possible to read and approximately locate the different architectural assets found in every municipality of the study area. Another major limitation in the process was the need to check some partial information against local historical archives.

#### 3.1.2. Historical Archives

Historical local archives are in two villages in the area Culla and Ludiente, where researchers have access to centuries-old data. The main limitation of these archives is the lack of complete and organised data due to loss, deterioration, or even the lack of classification. However, the advantage of accessing these sources is that as many local historians or academics from universities have already worked on them specific data can be easily retrieved by consulting publications. Thanks to these archives, it was possible to corroborate some toponymic information and to contrast and verify the information from the historical cartography.

### 3.1.3. Participation

Citizen participation is another fundamental pillar of preliminary works in this area [37]. The main participatory tool was meetings which were held, showing and explaining maps and other relevant information to local residents. We wanted to share and learn more about the cultural and natural values, for example the architectural–ethnographic essential elements seen in the historical cartography and mentioned in the archives and to corroborate and expand the initial information with that provided by the residents in the individual municipalities.

### 3.1.4. Fieldwork

The three-person working team for the field trips was eventually made up of four people: two specialists in architectural heritage, an agricultural engineer, and a biologist. Occasionally, local residents joined these campaigns to help the team of researchers. The three main fields investigated in this analysis were the communication pathways, the architectural and ethnographic assets related to those infrastructures, and the natural heritage of the entire area.

#### Communication Itineraries

Before the fieldwork, the analysis of cartography and archives and citizen participation was fundamental to help clarify some confusing aspects of these routes. By overlaying the information with GIS mapping software, it was possible to draw the original layout that the fieldwork would later corroborate. The first problem arose when it became clear that some of the routes, which had long been disused, were in a poor state of conservation. Although most of these trails were formerly used for pilgrimage to a shrine, today, only five villages maintain this tradition.

The local population was consulted again for verification, correction, and approval from these first campaigns. This was later validated with a second data collection campaign. The 173 km of routes were covered on foot over 13 days between March and June 2016. The overall objective of this second approach was to georeference the itineraries using a global positioning system and to relate the exact location of the architectural and ethnographic elements that were missed in the first field trip and the most relevant sites of considerable natural value.

#### Cultural Heritage

For the inventorying work, researchers used a template that reflects the general data of the property: a coding system for the itinerary, by typology and order of appearance, toponymical name, location using cartographic coordinates, element typology, style, chronology, and essential parts. Location plans were used to identify these both graphically and photographically. The second part of the model file contains a description of the cultural element, its historical use, its apparent state of conservation, and its contemporary use. The model also reflects the main pathologies and the cataloguing criteria, types and degrees of protection, and the recommended interventions for both the property and its surroundings. The result of this cataloguing was 227 architectural and ethnographic elements distributed over an area of influence of 50 m on either side of the 173 km long route.

At the same time, the paved stretches of these routes were also catalogued and mapped. These patches normally correspond to very steep slopes of complex access and with drainage problems. Original settlers in the area paved them with the dry-stone technique. For classification purposes, researchers divided them into two categories: (1) spot paving, which consisted of areas where the pavement surface was not continuous, or (2) pavement sections where patches of dry stone were continuous for several metres. After noting the initial and final position through waypoints, the paved areas were photographed and classified for each point or section.

Natural Heritage

During fieldwork, the natural heritage team identified five representative habitats. Firstly, there are the scrublands, environments dominated by low or medium-high plant species, such as rockroses or heathers. There are faunae that use the thickets to build their nests (birds such as the wheatear) or to feed on seeds or fleshy fruits and a variety of insects. Secondly, rivers, both in the high mountains and in the middle and lower courses, are very changeable environments. The flow of water varies according to rainfall and melting snow. Both the rivers and their banks are home to a great variety of plant and animal species. In addition, some of them have seen their course modified by humans to obtain energy for different uses. This is also the case of the kermes oak and oak forests, which are dominated by kermes oak, oak, or gall oak, depending on the type of soil and altitude. Insects abound in these forests. There are wild boars (*Sus scrofa*) that feed on acorns, birds capturing insects, and even wild cats (*Felis sylvestris*). Along with these, in the forests the most frequently found trees are Aleppo or Scots pines, on whose upper branches mistletoe, a parasitic plant with juicy fruits, may grow. At lower altitudes there are thorny and aromatic shrubs, some with fruits popular among animals, such as rose hips and hawthorn. Finally, the high mountain, a very harsh environment, where sparse and very low vegetation, such as grasslands or pastures, develops. Although few plants and animals are well adapted to these requirements, in Penyagolosa we can find the Alpine Accentor (*Prunella collaris*) or the mountain goat (*Capra pyrenaica*).

In terms of the anthropic occupation of the area it is worth mentioning farmhouses and their terraced plots as natural ecosystems. The orchards provided food for the Ortolan (*Emberiza hortulana*), in the wheat fields there were Corn Buntings (*Miliaria calandra*). The Long-tailed Tit (*Aegithalos caudatus*) builds its nest with the hairs of large mammals, such as sheep wool or spider webs and mosses. Hose Sparrows (*Passer domesticus*) feed on the remains of grain from the harvest and from chicken food. Geckos (*Tarentola mauritanica*) are found on farmhouse walls and Hoopoes (*Upupa epops*) nest in holes in farmhouse walls. In the roof and between tiles we can still find the lesser Horseshoe Bat (*Rhinolophus hipposideros*), an endangered species in the Valencia Region, while Savi's Pipistrelle Bat (*Hipsugo savii*) takes its refuge in small cracks as the stone walls of farmyards. The Iberian Lizard (*Podarcis hispanicus*) can be found near the walls of the farmhouse or in the lower parts of other constructions. In the watering zones for the cattle, there were *Gallipatos* or Spanish Ribbed Newts (*Pleurodeles waltl*), amphibians which can measure up to 30 cm, while the black Wheatear (*Oenanthe leucura*), a very striking bird with a black body and white tail, can still be found breeding on walls or in abandoned buildings.

3.1.5. Open Databases

Researchers employed open or freely accessible cartographic resources for this study [28]. These served both as an initial source of data to organise the information in both observation and cataloguing and to help outline the final methodology of the work. Cartographic data were obtained from the Cartographic Institute of Valencia (ICV), a scientific and technical body of the Regional Government of Valencia. From the ICV, researchers obtained the shapefile format layers of the municipal boundaries and data on protected areas (SCI, SPA, and N2000), as well as the types of soil and datasets produced within the framework of the Corine Land Cover programme, which refers to the state of land cover at European level. Similarly, the digital elevation models (DEMs) were retrieved to provide the altimetric data for the research through the raster cell-data format.

In addition, through the CNIG, laser imaging detection and ranging (LiDAR) researchers collected data to create more defined elevation models from those provided by the ICV. This helped in the observation and definition of anthropic elements and low, medium, or high vegetation areas through internal classification. Finally, the shapefile layers of the Habitats of Community Interest Directive of Annex I of Directive 92/43/EEC were used to superimpose these layers of the areas under protection onto the historical cartographic material and on-site data.

*3.2. GIS Work*

3.2.1. Data Collection

This is where the parallel lines of research converge. An exhaustive review of all the data obtained in the previous stages of the study was carried out in order to compile all the geographic and cartographic information. The revision phase is necessary given the numerous limitations existing. The integrity of the open data has to be questioned and an exhaustive comparison of the open, on-site, and old-map-related data needs is necessary.

The next phase included the data obtained from the revision of the planimetry. Two software programs were used for this purpose, with researchers using QGIS 3.16 and ArcGIS Pro 2.9.0 to facilitate data manipulation. Each process resulted in different projects with vector layers using points, determined by elements whose relevance relied on their location; lines or elements whose main attribute is distance; and polygons, auxiliary elements for the interpretation of the zoning.

3.2.2. Classification of Fieldwork Data

Creating a fieldwork database relies on the cataloguing process and the interpretation of results. To ensure no information is missed, each field within the attribute tables was completed with brief but sufficient information on each element. An element code, name, and coordinates in ETRS89 UTM zone 30N, altitude, village, type, subtype 1, and subtype 2 were used.

This information was used to map and analyse the potential linkage of elements to the itineraries. Although an attempt was made to contrast our data with classifications obtained from open sources, some results did not overlap within a specific distance, under 50 m. Similarly, there were toponymical mismatches and after corroboration with bibliography, citizen participation, archive work, and historical cartography, researchers decided to follow the local popular toponymy.

Apart from the itineraries, other types of cultural assets specifically included:

- Civil architecture: this includes farmhouses and surrounding auxiliary constructions such as farm buildings, or others such as ovens, rural schools, and shepherds' shelters or huts.
- Hydraulic architecture: this group is directly related to water use and extraction and includes wells, waterwheels, fountains, mills, dams, water ponds, and snow pits.
- Religious architecture: this category is directly related to worship and includes hermitages and churches.

To enhance the classification of data, researchers created a rubric to qualify the importance of each element. The fieldwork team fulfilled different rubrics according to types and subtypes of cultural elements. Technicians valued the assets both for their uniqueness and state of conservation. Distance and accessibility were also evaluated considering the road or pedestrian access and the type of road surface. Finally, the social value of the elements was an additional reference obtained from the meetings with the population.

Researchers ranked the relevance of elements according to three levels. "Good" is assigned when the average value of the element to be preserved is in the final third of a punctuation scale out of 100, "Regular", when the value reaches the second third, and "Deficient" when the value reaches only the first third.

3.2.3. Relationship between Chosen Factors

The analysis of relationships is fundamental to the final decision-making stage. Therefore, using the open data layers and those resulting from the fieldwork, researchers generated a new table of attributes in which categories were associated with geographical positioning. This process is cumulative: Figure 3 details the process of adding layers to make decisions based on the qualitative values generated, their positioning, and the value of the environment until the most significant elements are reached.

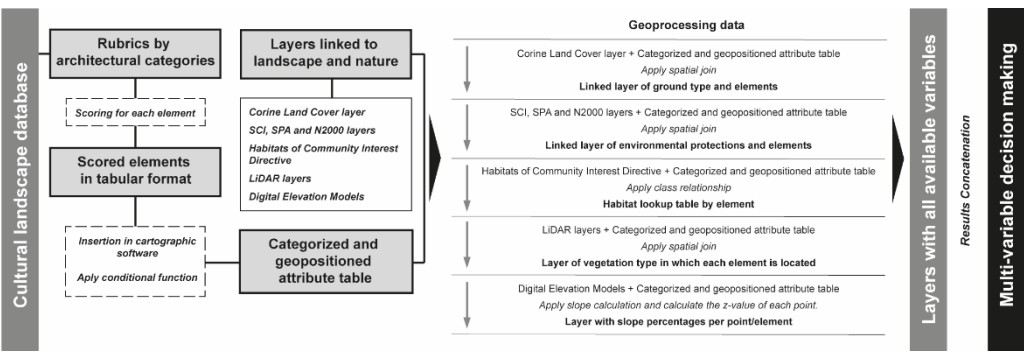

**Figure 3.** Description of the geoprocesses used in the methodology of the study.

The integration of the rubric into the software and the previous calculations were performed in tabular format and integrated into the attribute table of the element layer from the coding and toponymy initially created in the cataloguing. From there and using the different variables, a logical if-else function was created to implement case change statements. This function aims to categorise according to a value, in this case that of the rubric, giving different ranks according to their score. At this point they started to categorise and prioritise according to their score and the outcome of the geoprocessing.

Researchers solved the relationship between cultural and natural heritage using the cartographic software together with a spatial join geoprocessing tool. The entity's attributes created in the cataloguing phase were merged with the Corine Land Cover layer. By default, all the attributes of the joint entities were incorporated into the attributes of settings and itineraries and copied to the output entity class, so after each of these operations, the resulting attribute table was simplified to allow the essential defining values to appear. This relationship arises from the geopositioning of each element. By overlapping layers in polygon and point format, they can be associated and linked through their attribute table. Thanks to this, the relationship between the elements and the predominance of soil type is established. Given the complexity of the habitat layer, which is an overlapping polygonal layer where the same natural type of environment can have several specifications, the creation of a class relationship was decided to store an association between the features of fields in the source table (habitats) and the target table of elements' layer. Researchers employed the Create relationship class command through the geoprocessing tools. This query tool selects elements in order to find out their positioning in a specific habitat. In this context, LiDAR layers were used to identify the type of vegetation in each setting. The initial laz format was changed to a format compatible with the software and a database was created using the Create LAS Dataset function. The result was a database where the 105 initial laz layers could be managed immediately. Given the density of points where each element was located, it was possible to see the state of the vegetation where it was located by differentiating four categories: high, medium, low, or non-existent. The final function of this process was to concatenate or join all the columns resulting from the analysis into one where all the data obtained could be seen in order to start the analysis both in tabular and cartographic format through categorisations and symbology.

For the calculation of slopes and their relation to the pavements of itineraries, usually associated with rough terrain or with drainage problems, researchers needed to know the slope where the paved areas are located. The geoprocessing tool used calculates slopes in percentages from a DEM. With this new slope surface, another geoprocessing tool was employed to calculate the z-value (in this case, the value of the slope pixel) to be added to the points where there are paved segments through a spatial union.

## 4. Results

The principal value of these results is to create a verified starting point from which to access a verified database on this region's built heritage and natural environment. Its

interpretation, in a staggered manner, adding the appropriate characteristics in each case, will serve to refine a management plan and the prioritisation of elements.

### 4.1. Results of Fieldwork

#### 4.1.1. Abundance and Distribution Analysis of Natural Elements

The areas signified by the Nature 2000 Network, SCI (from habitats directive), and SPA (birds directive) were retrieved as open data. Figure 4 shows their distribution in the area under analysis, most of which is under SCI, SPA, or N2000 protection. Management regulations, apart from those of the Natural Park, must include conservation measures to maintain or achieve a favourable preservation status for the habitats and species that led to the inclusion of these areas in the N2000. In addition, issuing mandatory management regulations makes the eventual conversion of the SCI into Special Conservation Areas possible. Among the general regulations already existing in the area are those relating to the public hydraulic domain; to soils and geological resources; to wild flora and fauna; and to cultural heritage. However, there is no umbrella policy or plan to connect all values and relevance. A first advance proposed by this research is to gather all these areas in a single database to avoid omission in the event of evaluation of the territory. Linking these protections to an element is a step forward in the study of this territory.

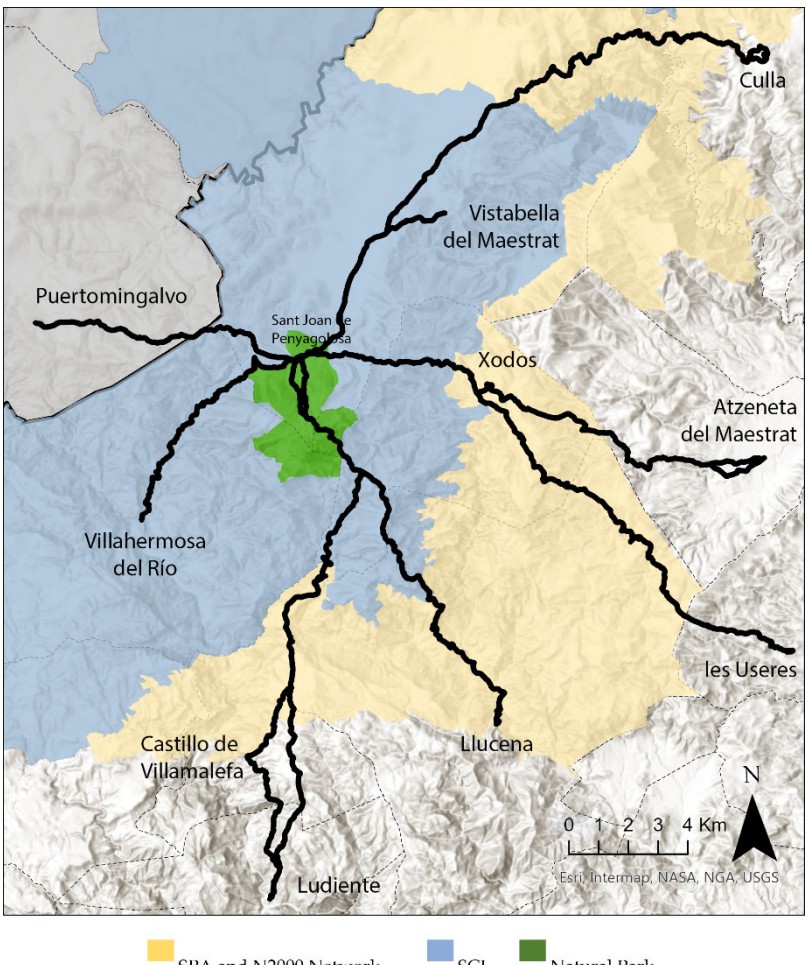

**Figure 4.** Distribution of areas protected by environmental and biodiversity legislation. Source: authors.

The value of LiDAR in this study relied on observing and understanding the spread of vegetation over pieces of land that were previously occupied by humans through the analysis of vegetation layers, low, medium, and high vegetation density. The study provides relevant insights that were integrated in the valuation of settings and itineraries.

Figure 5 shows a three-dimensional photographic image where vegetation encroaches upon past agro-pastoral landscapes. LiDAR helped researchers visualise this effect by filtering vegetation as point clouds via satellite images. The data obtained from LiDAR are current as no earlier data are available. However, all analyses require a starting point and initial monitoring. The coincidence between the abandoned architectural elements and the higher density of vegetation points is also noteworthy. With sufficient data contrast, this gives a proportional and direct relationship between the location of construction and the state of its surroundings when cartographically, it can be confirmed as a cultivated area (either by the Corine layer or by the morphology of the terrain) and its state of conservation is specified.

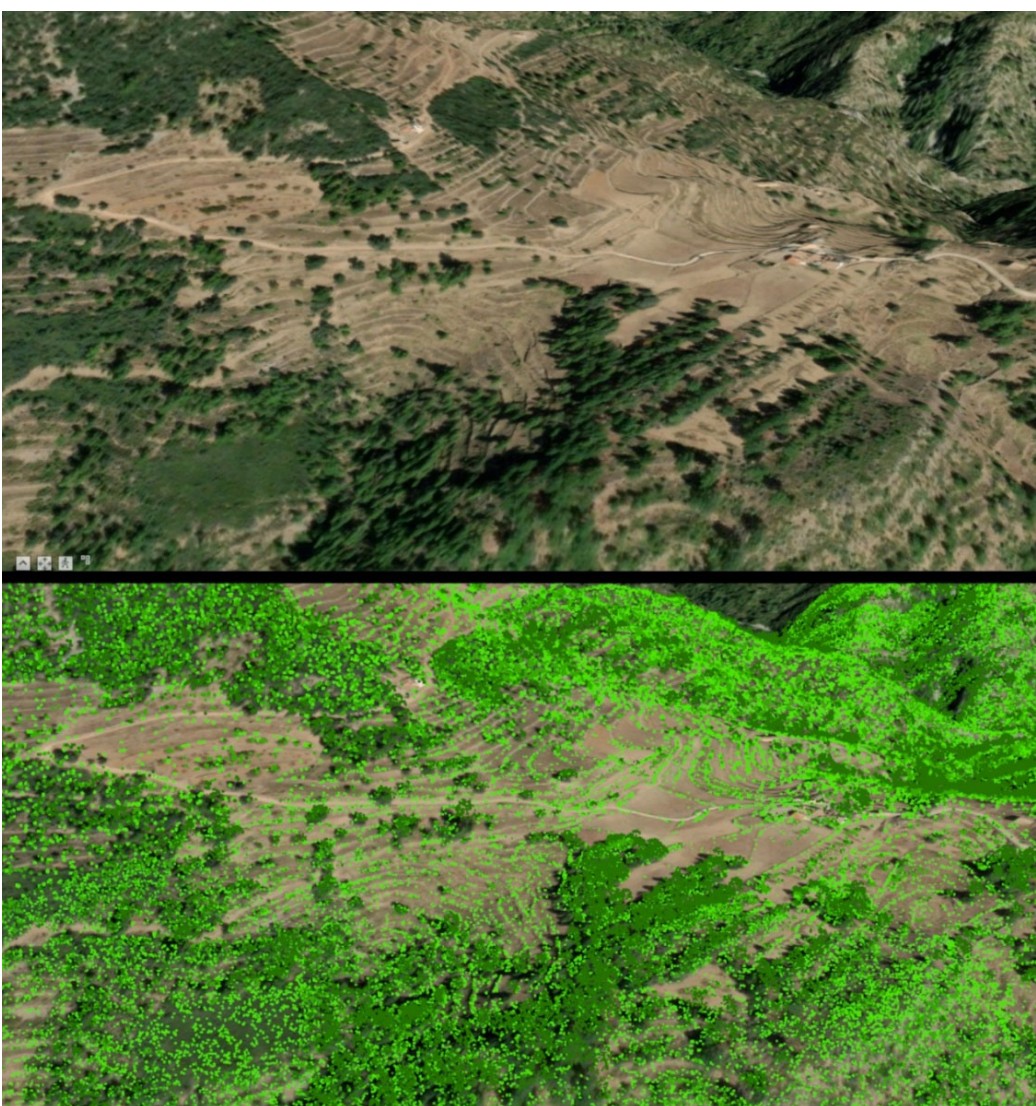

**Figure 5.** Orthophoto and LiDAR point clouds used for vegetation analysis. Source: authors.

### 4.1.2. Abundance and Distribution Analysis of Cultural Elements

Fieldwork provided us with several elements scattered in the landscape. The information necessary to define and classify these was backed by the existing levels of protection. The information is summarised in Table 1, which shows the distribution of elements by type. Farmhouses stand out with 121 elements in very different states of conservation. There is a significant number of water-related elements (68), which are predominantly associated with farmhouses. The number of religious assets (20) is also considerable. While there is one church per village, this is not the case with hermitages, being less than one per village.

**Table 1.** Distribution of the cataloguing of the Penyagolosa area according to types and subtypes of elements.

| Type | Subtype 1 | Subtype 2 | Count |
|---|---|---|---|
| Civil | Farmhouse | | 121 |
| Civil | Archaeology | | 1 |
| Civil | Barn | | 2 |
| Civil | | | 1 |
| Civil | Kiln | | 4 |
| Civil | Defensive | | 1 |
| Civil | Farmhouse | Defensive | 2 |
| Civil | Rural school | | 2 |
| Civil | Shelter | | 2 |
| Hydraulic | Well | | 9 |
| Hydraulic | Waterwheel | | 8 |
| Hydraulic | Fountain | | 27 |
| Hydraulic | Mill | | 7 |
| Hydraulic | Weir | | 8 |
| Hydraulic | Water rafting | | 6 |
| Hydraulic | Trough | | 1 |
| Hydraulic | Snow pits | | 2 |
| Religious | Church | | 10 |
| Religious | Hermitage | | 9 |
| Religious | Hermitage | Defensive | 1 |

A planimetry was generated where each point is located within the corresponding subtype from the layer of point entities. For example, as can be seen in Figure 6, the researchers included waterwheels, perhaps a very residual element of the landscape but a very significant one. These elements belong to specific locations, mainly basin or flat areas, to extract underground water [6,38].

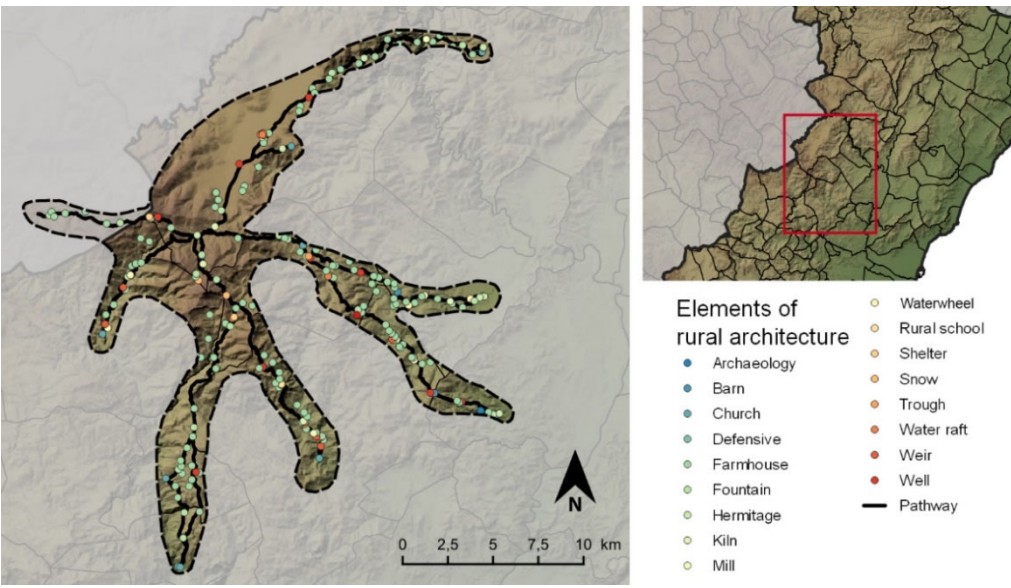

**Figure 6.** Planimetry of the Penyagolosa area with the georeferenced and categorised elements.

### 4.1.3. Paving

The itineraries' stonework was considered of sufficient quality for inclusion in the cataloguing process. Figure 7 shows a general overview of the 281 sections of varying quality and state of conservation, which the researchers catalogued through fieldwork.

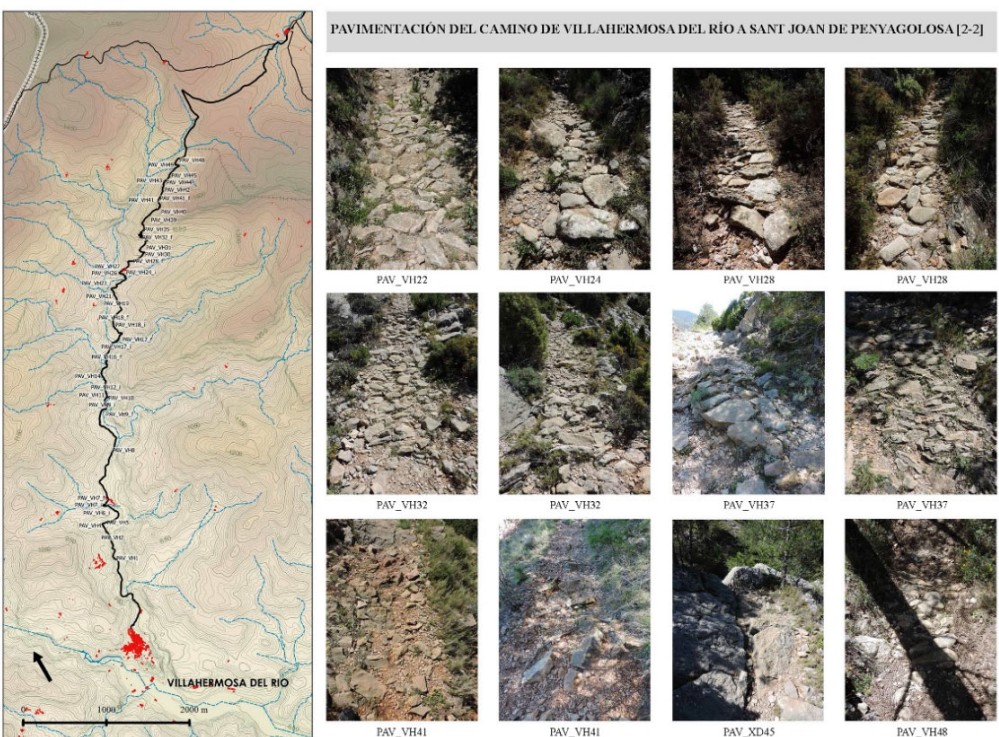

**Figure 7.** Descriptive card of the cataloguing of pavements. The sheet included a plan with the catalogued points and images of the areas with their cataloguing code.

The analysis of slopes using ArcGIS Pro allowed the researchers to retrieve the location of the stone-covered surfaces spatially and relate them with the percentage of slope in every itinerary. To sum up, the average slope of the paved areas is 32.14%, while that of the maximum slope is 69.08% in a continuous section, dropping to 21% in the final section. This demonstrates the relevance of the analysis for future preservation; otherwise, the continuous loss of itineraries is guaranteed. Figure 7 shows the degradation of some stretches.

*4.2. Landscape-Related Results*

4.2.1. Classification of Cultural Asset Entities and Proposal for Intervention

As explained in Figure 3 and in Section 3.2 of the Methodology, Figure 8 shows the distribution of some of the catalogued elements according to their state of conservation and the priority of intervention. After filtering, in this experimental methodology, a total of 31 elements was selected based on the three main criteria (natural, cultural, and social).

To classify and map the elements, the rubric layer for their conservation state and the one for intervention priorities were cross-referenced. The importance of intervention was prioritised according to the relevance of cultural elements and the natural area surrounding them. Itineraries and settings with high biodiversity in an outstanding natural environment were valued as high priority. Medium priority was attached to itineraries and settings with high biodiversity in an environment that is not outstanding in social, cultural, or natural terms. Finally, low priority was assigned to the rest of the assets. Researchers added all these valuation remarks as an attribute table in ArcGIS Pro software.

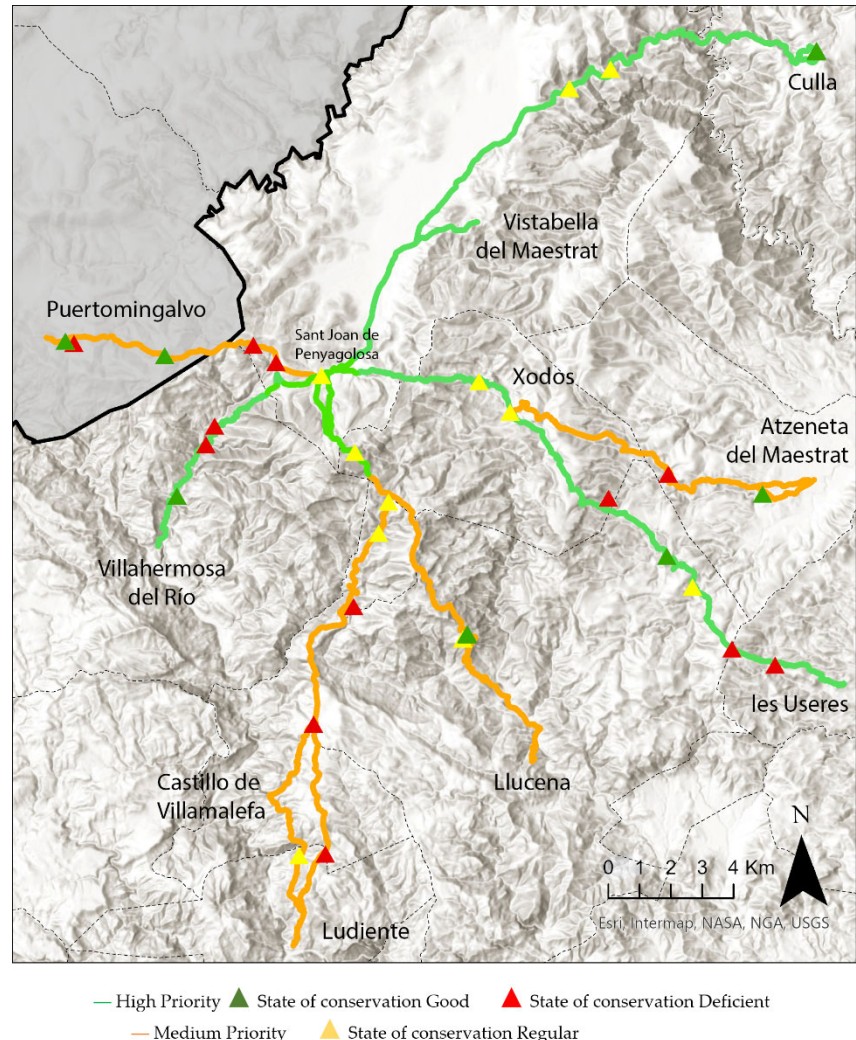

**Figure 8.** The prioritised itineraries are: medium (orange) and high (green), and the state of conservation of the other cultural assets is: good (green), regular (yellow), and deficient (red).

The links between the two layers provided the final results of the two attribute tables, allowing researchers to put forward a coding system to be used for an intervention proposal as part of an integral plan of management. The intervention proposal is subdivided into two. The first-order interventions consist of carrying out consolidation or repair works to prevent the structural collapse of cultural elements in a poor state of conservation. The second-order interventions relate to non-structural elements.

When intervention is not a priority (green), the conservation objective is focused on maintaining the current state. When it is considered Regular and Deficient, the conservation objective is to maintain the cultural asset in the same state of conservation or to try to improve the element with actions that do not affect its integrity or authenticity. Once the priority of intervention and the level of conservation are assigned for each element, in line with regional policies, the management plan proposes the frequency of actions according to intervention priority and conservation level.

### 4.2.2. Natural and Cultural Heritage Settings and Their Role in the Landscape

The steps carried out in the methodology applying quantitative data to the study territory—and the response to certain data given that this is a constructed landscape—can be subject to a qualitative evaluation and linked to the first (social) steps of the research. To establish the protection and value criteria for the elements, the following were weighted: (1) the possible proximity of catalogued birdlife or birdlife of interest depending on the

existence or not of suitable habitats nearby; (2) the accessibility to the cultural and natural setting; and finally, (3) the potential effect of tourism, as well as the occasional pressure of visitors throughout the year. The application of these three criteria resulted in four types of settings, classified according to their state of conservation and potential threats. Researchers explain the assessment process taking those from the Les Useres itinerary as an example (Figure 9).

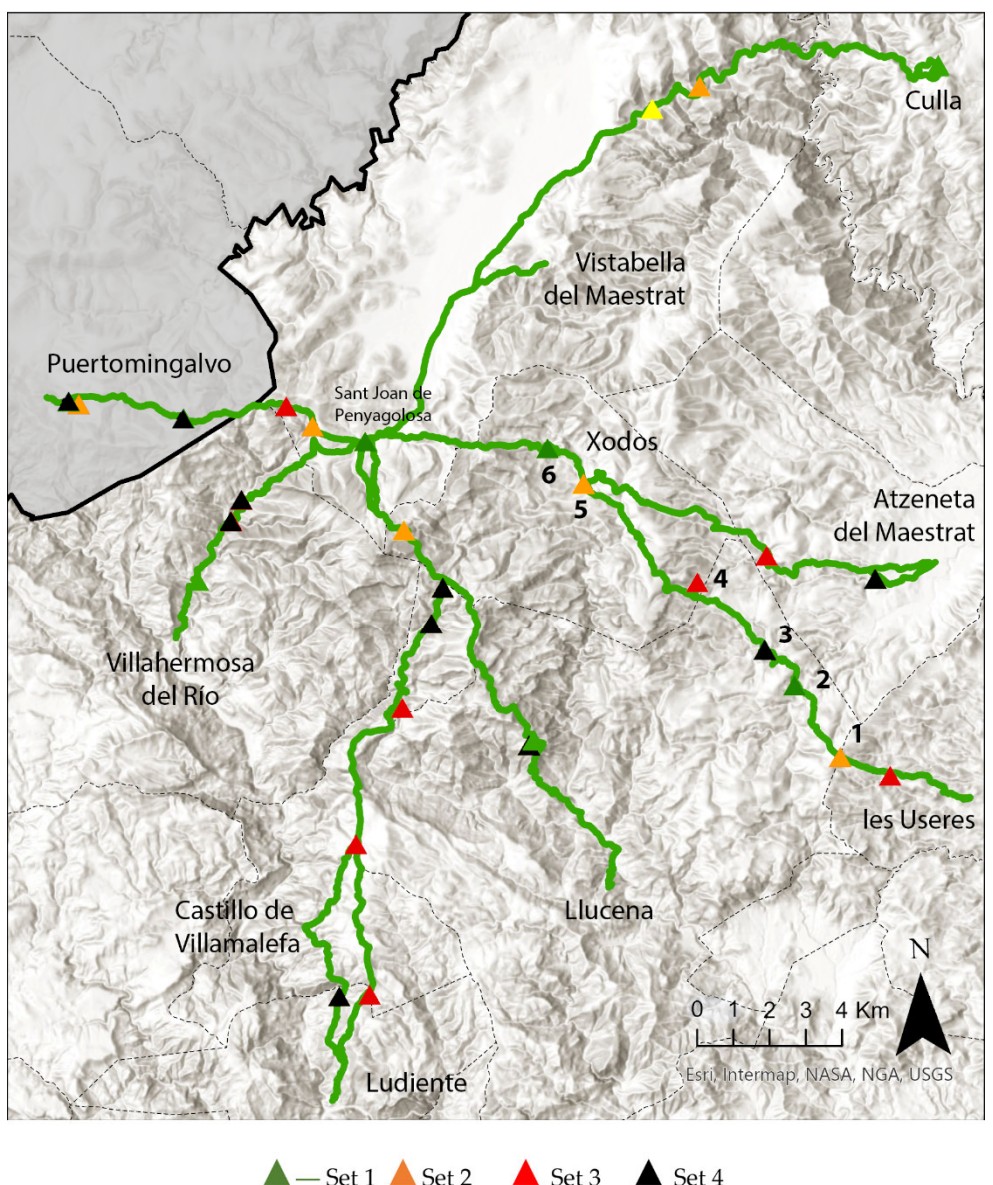

**Figure 9.** Settings of cultural and natural value. Source: authors.

The first type of settings for elements to be preserved should be considered a priority. In this case, as in the rest of the itineraries, the paving stonework is tremendously fragile. Among the architectural elements ranked as first priority, the farmhouse Mas de Blai de la Vall (2) stands out, despite being in a moderate state of conservation. In an intermediate section of the pathway, the mill Molí de Xodos (6) is also relevant and in a relatively good state of preservation. Following the recommendations of the Plan, the use and enjoyment of these elements should be carried out at times that are not sensitive to the fauna, i.e., outside the breeding period. If action is taken on water elements to favour the presence and flow of water, the laying and growth periods of amphibians should be respected to provide escape structures for the fauna. In this way, impacts on the natural heritage can be minimised.

The second type of setting includes the farmyard Corral Blanc (1), at the beginning of the itinerary, which is in a poor state of conservation. There is also the Mas de la Vega farmhouse (5) in Xodos, which is in a moderate state of conservation. Prior to any possible interventions in these sites, fauna and flora studies are required to ensure that listed species are not affected. Without contravening the natural heritage regulations, work on cultural assets can be carried out outside the reproductive periods of fauna. However, a prior inspection of sites must be carried out to certify the absence of fauna using these environments. As these settings are located near environments of interest, some listed species, such as the Short-toed snake Eagle (*Circaetus gallicus*), Montagu's Harrier (*Circus pygargus*), Peregrine Falcon (*Falco peregrinus*), or Ortolan, (*Emberiza hortulana*), could be present. In this case, the timing for interventions or maintenance actions should be carefully evaluated.

In the third type of settings, the defensive hermitage of Sant Miquel de les Torrocelles (3) is catalogued as being in a good state of conservation. However, these types of settings are in very fragile environments. In the event of possible actions on the elements or in their immediate surroundings, the nesting fauna and flora should be respected in order to enhance the natural value of the area. Therefore, in these settings, changing the use of the heritage assemblage is not advised in order to avoid a high influx of visitors that could worsen the state of conservation, potentially leading to disturbances to different natural species. Without contravening the applicable regulations, restoration or tourist activities can be carried out if impact minimisation measures are applied and sensitive periods, both for birds and other vertebrates, are respected. In addition, it is necessary to assess whether the actions can potentially attract more visitors and which of these could pose a future threat and so, be inadvisable.

The fourth type of setting from this itinerary includes the Mas d'Ahicart de Dalt farmhouse (4). As with other elements from the rest of the itineraries, it is recommended that no action at all is taken on these settings, either because of the fragile surroundings or because of the inaccessibility to the cultural heritage elements. However, any conservation action would increase the impact on the natural value of the site by creating new operative accesses. Therefore, these settings require critical decisions as regards accepting cultural abandonment in favour of enhancing biodiversity.

### 4.2.3. Implications

The entire methodological process attempts to create management directives applicable to vernacular landscapes. Based on this study, six lines of action could be useful in the further research on actions to preserve the cultural and natural heritage of similar areas:

- First: Implement a coherent and integrated management plan for the natural and cultural resources according to local specificities and legislation.
- Second: Promote investment for the analysis of cultural and natural assets to ascertain the impact of rural abandonment on cultural and natural preservation.
- Third: Further investigate the way technology serves to collect and retrieve information to improve critical managerial points.
- Fourth: Implement a wide-ranging programme for the documentation and knowledge dissemination of the landscape and its relevance.
- Fifth: Undertake a feasible programme of interventions and maintenance of infrastructures, architecture, and ethnographic elements.
- Sixth: Implement a programme for the conservation and restoration of natural areas.

## 5. Discussion

The methodology developed in this article for creating heritage databases is suited to the analysis of all types of vernacular landscapes, from which cartographic data can be obtained. However, each environment requires specific consideration. In this case, in addition to the environment, the itineraries are the core element and the architectural and ethnographic assets act as a link between people and villages. Some of the elements the

researchers found in the landscape, such as the paving of the itineraries, are vulnerable and have no legislative protection. In addition, the system of settings was created to carry out respectful interventions in both the cultural and natural values expressed by residents in the preliminary meetings. Therefore, the configuration of these settings was guided by a balanced mix of actions and interventions.

One of the most critical points of the project—and its main limitation—was the collection of information. As seen throughout the methodology, the article uses a wide variety of sources and fragments to compile the location, toponymy, and use of different agricultural spaces, habitats, and constructions. In this sense, when working in environments with small populations, the information is usually hidden. It is unusual to find thematic cartography, official databases based on geopositioning without excessive margins of error or habitat positioning with specific delimitations and sufficient attributes to carry out an in-depth analysis. Furthermore, although some of the works mentioned imply a direct link between habitat and built space, they are few and far between and tend to focus on one of the two variables. Consequently, it was inevitable that the initial quantitative work should become qualitative in the final decision-making stage and in the implications derived from the analysis. The third and final limitation derives from the results as this work is not intended as a canon on conservation. The qualitative response to quantified data is the result of reflecting variables that can aid in the respectful preservation of both the built environment and the space where it is located.

To overcome these limitations, it was decided to implement the maximum number of variables in the process. As Hearn [38] explains, the solid ethnographic component of these studies gives rise to more questions, broadening the scope of the research. Although complex because of the difficulty in obtaining data, expanding the research resources to the maximum available provided a good starting point for the project. In fact, as Fredheim and Khalaf [39] explain, meaning often goes beyond the material. In this case, local people still preserve and recreate most cultural and natural assets, detailing how attributes and values are maintained and transmitted over generations. Consequently, as also suggested by Beel et al. [40], the involvement of residents helped us to clearly describe the relationships and connections that continue to maintain the character of the vernacular landscape today. The basic idea of preservation and resilient management is reflected in the concepts of esteem and place attachment [41]. Although the environment is limited and of a small territorial scale, the use of multiple variables in decision-making should be appreciated.

Although the evaluation and cataloguing of landscape heritage require a rigorous methodology procedure, in this case, databases can implement soft values to those that are more clearly defined. In landscape assessment, the term soft value is used to describe the qualities of diversity, sensitivity, perception, identification, and value, attached to both the tangible and intangible assessment, such as, for example, the sense of place or community factors that could not be identified exclusively through technical work. Comprehensive management plans are an excellent tool for reflecting on the protection model. They allow us to move decisively towards a new culture of land-use planning that involves the stakeholders' approach, knowledge and feelings, and the integration of cross-cutting figures.

In this sense, the study demonstrates that a detailed and multidisciplinary categorisation of a large territory, with many natural and cultural resources, can be successfully assessed. The study demonstrates that there are different areas and assets with peculiarities and relevance for preservation. The strength of a joint natural and cultural assessment proves that comprehensive conservation is possible. However, the research methodology tested here is applied at local scale and depends on the stakeholders involved. Nonetheless, the systematic assessment of vernacular landscapes is useful in ensuring action protocols to inform administrations and legislation.

## 6. Conclusions

This work is part of an experimental study that considers cultural and natural assets in a vernacular landscape. The connection between the natural and cultural values of the landscape is one of the central values and innovations of this study. Specific protections for the study area derive from cultural heritage regulations and from natural areas, landscape, and biodiversity regulations. The combination of these two fields of inquiry in a methodological approach is an unusual practice of further use in future assessments.

The challenge of this study is to update the data. It should be taken into account that we are starting from a universe with scattered, unconnected data, which need to be united in order to begin to understand the territory from an analytical point of view. Once this joint vision of the natural and built environment is created, the next step is to update the data through monitoring to continue investigating the state of conservation of the landscape. Subsequently, databases can be created that are updated at pre-established intervals to obtain a weighted view of the changes in the landscape.

LiDAR images, together with historical information and fieldwork data, demonstrate the dual character of the landscape: natural and anthropic. The analysis shows how hundreds of species in the area of study need both anthropised and wild habitats to be preserved. The study has also highlighted how interdependent architecture and nature are, as the living spaces of humans and animals are now shared. The same happens in the troughs around the farmhouses, where animal survival is linked to the maintenance of good practices on the livestock and farming activities. These interactions have been recognised as settings of particular interest.

One of the priorities of this document is to recognise, promote, and develop a methodology to safeguard the diversity of vernacular landscapes, with culture and nature as characteristic elements. The tools we tested function as a practical approach at the detection level. However, these tools are exploratory, given their potential to develop further research and educational procedures. Within the evaluation process, there were many essential steps to effectively inform conservation and intervention issues. Through this method, the research offers an alternative, and perhaps more coherent, perspective to enrich the dialogue and ensure integrated valuations focused on the social, natural, and cultural realm.

One of the strengths of this study is its multidisciplinary perspective, as the multifocal approach considers stakeholders to be the main sources of information and valuation. Therefore, this research is helpful, first and foremost, in the sharing and creation of knowledge, as well as in terms of integration and as a technology-based management tool for soft value cataloguing.

Finally, it should be added that this study is currently being expanded to other neighbouring regions similar to the current study area, but with their own specific particularities. This methodology can be applied effectively and approaches the landscape from a constructive and positive perspective. It is affordable enough for any administration to be able to perform a landscape diagnosis. As the study is replicable in any agropastoral landscape, it is necessary to ascertain the maximum number of variables to create a universe with all the possible variables, as stated earlier. Given that replicating the experiment is expected to lead to updated technology as well as data, in the future, it will be possible to improve and streamline the methodology, making it easier to extrapolate to other territories.

**Author Contributions:** Conceptualization, J.A.G.-E.; methodology, J.A.G.-E. and P.A.; software, P.A.; validation, J.A.G.-E., A.V. and P.A.; formal analysis, J.A.G.-E., A.V. and P.A.; investigation, J.A.G.-E., A.V. and P.A.; resources, J.A.G.-E., A.V. and P.A.; data curation, J.A.G.-E., A.V. and P.A.; writing—original draft preparation, J.A.G.-E., A.V. and P.A.; writing—review and editing, J.A.G.-E., A.V. and P.A.; visualization, J.A.G.-E., A.V. and P.A.; supervision, J.A.G.-E. and A.V.; project administration, J.A.G.-E.; funding acquisition, J.A.G.-E. All authors have read and agreed to the published version of the manuscript.

**Funding:** This research was funded by Ministerio de Ciencia, Innovación y Universidades: MCIN/AEI /10.13039/501100011033 [grant number PID2019-105197RA-I00]. Pablo Altaba was funded by the postdoctoral programme PINV2020-Universitat Jaume I [POSDOC/2020/06].

**Data Availability Statement:** Not applicable.

**Acknowledgments:** The authors want to thank all the volunteers that participated in the project, the local administrations, and Belén López Precioso for the legislative advice.

**Conflicts of Interest:** The authors declare no conflict of interest.

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
