# Peer review of "Assembling Cultural and Natural Values in Vernacular Landscapes: An Experimental Analysis"

_remotesensing, doi:10.3390/rs14174155_

Round 1

Reviewer 1 Report

The work has a good starting point in the analysis of the cultural landscape. It proposes a broad methodology that includes not only the use of old maps, open digital datasets, and media applications through GIS. Fundamentally, he considers public participation in mapping for value elicitation to be an essential part of the process. This allows authors to present the results and sets out the starting points and limitations for a conservation plan.

Author Response

The work has a good starting point in the analysis of the cultural landscape. It proposes a broad methodology that includes not only the use of old maps, open digital datasets, and media applications through GIS. Fundamentally, he considers public participation in mapping for value elicitation to be an essential part of the process. This allows authors to present the results and sets out the starting points and limitations for a conservation plan.

Firstly, we would like to thank the  reviewer for their comments, which will undoubtedly help improve the article. Below, we answer the suggestions provided and explain where modifications have been added to the text. In addition, the paper has been revised to improve the language.

We are working on an applicable methodology. We are also optimistic about the conservation of environments that are now vulnerable. This shows that it is possible for local projects to be created through involvement and collaboration.

Reviewer 2 Report

A GIS-based approach aimed to characterize and classify vernacular landscapes is proposed.

This study is interesting, but it is necessary that the authors highlight in the introductory section the added value of this research with respect to the GIS-based approaches proposed in the literature for the detection and classification of cultural and natural elements. What are the limits of this research and what processes are used to overcome them?

In section 3 it is necessary to add a description structured by steps of the experimented approach; it would be optimal to insert a figure that shows the flow diagrams of these steps. 

Paragraph 3.2 needs to be better structured. The GIS tools used (ArcGIS Pro, Quantum GIS) must be indicated only once, while for the description of the experimented approach it is necessary to use a more structured method. For example, instead of reporting "The following function was used: IIF ($ feature.Score> = 67," Good ", IIF ($ feature.Score> = 33 && $ feature.Score <67, "Regular", "Deficient")) ", it is necessary to specify the reasons for this rule implemented with IF statement, the meaning of the variables contained and the meaning of the values used for these variables.

What are the future prospects? Do the authors plan to carry out further experiments in different contexts? Do they intend to improve / optimize the proven processes in the future? These considerations should be added in section 6.

Author Response

Firstly, we would like to thank the reviewer for their comments, which will undoubtedly help improve the article. Below, we answer the suggestions provided and explain where modifications have been added to the text. In addition, the paper has been revised to improve the language.

A GIS-based approach aimed to characterize and classify vernacular landscapes is proposed.

This study is interesting, but it is necessary that the authors highlight in the introductory section the added value of this research with respect to the GIS-based approaches proposed in the literature for the detection and classification of cultural and natural elements. What are the limits of this research and what processes are used to overcome them?

The corresponding explanation has been added in the last paragraph of the introduction and in the discussion. On the one hand, the added value is to create a starting point for analysis and combine it with the legislative aspect to reach a global understanding of the environment. On the other, this also prompts discussion on  the limitations of a study of an unpopulated region with an abandoned landscape: lack of verified data, disconnection between these, etc.

In section 3 it is necessary to add a description structured by steps of the experimented approach; it would be optimal to insert a figure that shows the flow diagrams of these steps. Paragraph 3.2 needs to be better structured. The GIS tools used (ArcGIS Pro, Quantum GIS) must be indicated only once, while for the description of the experimented approach it is necessary to use a more structured method. For example, instead of reporting "The following function was used: IIF ($ feature.Score> = 67," Good ", IIF ($ feature.Score> = 33 && $ feature.Score <67, "Regular", "Deficient")) ", it is necessary to specify the reasons for this rule implemented with IF statement, the meaning of the variables contained and the meaning of the values used for these variables.

In addition to the modified text in section 3, figure 3 has been incorporated, with a concrete summary using a flow chart of all the steps followed to combine the layers examined in the article.

What are the future prospects? Do the authors plan to carry out further experiments in different contexts? Do they intend to improve / optimize the proven processes in the future? These considerations should be added in section 6.

Future lines of work for this method have been added to the conclusions. In these, we discuss the avenues of expansion to other neighbouring regions with unique particularities, similar to those of the current study area. We commented on the actual value of the research, as we believe this methodology is truly applicable and will have a constructive and optimistic effect on the landscape. Furthermore, its affordability will allow different administrations to perform a landscape diagnosis. We consider that the study is replicable to any agropastoral landscape. However, it is necessary to know the maximum number of variables in order to create a universe with all possible variables. Replicating the experiment requires updating the technology, not only the data, so that the methodology can hopefully be improved and streamlined to make it easier to extrapolate to other territories.

Reviewer 3 Report

the manuscript lacks original features. as much as the authors may have described the project carried out, no significant results were shown.

The authors should completely revise the article to give it a more scientific character. As it stands, the manuscript lacks original features and should possibly through figures and processes make the intended design more evident.

Author Response

Firstly, we would like to thank the reviewer for their comments, which will undoubtedly help improve the article. Below, we answer the suggestions provided and explain where modifications have been added to the text. In addition, the paper has been revised to improve the language.

the manuscript lacks original features. as much as the authors may have described the project carried out, no significant results were shown.

The authors should completely revise the article to give it a more scientific character. As it stands, the manuscript lacks original features and should possibly through figures and processes make the intended design more evident.

The article examines an area that has been populated for over a thousand years and abandoned for about 70 years. Although the problems are detailed in the article, we would be keen to transmit accessible research to any group of researchers interested in the landscape. It should be understood that we start from a data gap. The combination of historical cartography, participation, open databases, and fieldwork is an original aspect of the article, as studies tend to focus on a single aspect. In addition, its scale also makes it possible for other administrations to use this methodology for landscape management.

In addition, the heritage elements are quantified, while their surroundings, the natural part of the landscape, are linked to them. In this Mediterranean region, no similar study has been found from which to start at this scale and with these variables. For a better understanding of the article, the methodology has been redesigned by adding flow diagrams that transmit the concrete steps of the decision-making process.

Reviewer 4 Report

Figure 1 shows no longitude and latitude coordinates at the corners.

The Lidar data is an essential one in this study, but there is no description in Figure 2. There are not enough explanations in the text.

Multivariate decision-making is a quantitative statistical technique, but there is no explanation.

It is not easy to find any academic novelty. Moreover, the interpretation of results seems not to be an objective approach. Justify it, please.

This study needs a quantitative approach rather than a qualitative interpretation.

Rather than Remote Sensing, it would be appropriate to contribute to IJGI journals or journals in the field of geography.

Author Response

Firstly, we would like to thank the reviewer for their comments, which will undoubtedly help improve the article. Below, we answer the suggestions provided and explain where modifications have been added to the text. In addition, the paper has been revised to improve the language.

Figure 1 shows no longitude and latitude coordinates at the corners.

Figure 1 has been improved thanks to the reviewer’s comment.

The Lidar data is an essential one in this study, but there is no description in Figure 2. There are not enough explanations in the text. Multivariate decision-making is a quantitative statistical technique, but there is no explanation.

To improve the understanding of figure 2 and to add all the variables of the geoprocess in order  to understand the decision-making process, figure 3 has been created to explain each analysis performed step by step.

It is not easy to find any academic novelty. Moreover, the interpretation of results seems not to be an objective approach. Justify it, please.

The academic novelty that we propose arises, on the one hand, from the territorial scale used, which is much smaller than that usually used in landscape analysis. On the other hand, the union of different types of data such as historical mapping, archives, public participation and open data as a universe can be evaluated and applied to a particular territory. Furthermore, the establishment of criteria for restoration in the function of the environment benefits all the heritage elements. It does not link the criteria to economic or situational factors but to the natural environment. In addition, as the implications are general and non-binding, they are not biased and do not point out or exercise value judgments.

This study needs a quantitative approach rather than a qualitative interpretation.

From the methodology used, it can be seen that quantifiable data are used for decision-making in the results phase. However, we consider that quantitative data must have a qualitative function in landscape issues. In addition, decision-making on a reduced scale cannot be only numerical. Qualitative territorial analysis is needed to understand a changing and adaptive environment and to propose nuanced solutions.

Rather than Remote Sensing, it would be appropriate to contribute to IJGI journals or journals in the field of geography.

We believe this journal is appropriate because we propose starting monitoring for a specific region. We talk about the importance of updating geopositioned data in a particular area. The initial absence of data is in itself a key point when submitting the article to this journal. However, we feel the results are promising enough to us to be able to establish a database that can be updated according to geospatial technologies.

Round 2

Reviewer 2 Report

Authors responded comprehensively to all comments, further improving the quality of their paper. I consider this paper publishable in the current form.

Author Response

Thank you very much for these and previous comments. They have been of great help in improving the work. Thank you very much for your work.

Reviewer 4 Report

Is the period in the title appropriate?

Assembling cultural and natural values in vernacular landscapes: An experimental analysis

Technical descriptions of Fig. 3 are still insufficient.

Author Response

Thank you very much for these and previous comments. They have been of great help in improving the work.

Regarding the first comment, we consider the proposed title appropriate as it speaks of "Assembling ", which is the main objective of the article in terms of natural and cultural values, and of "experimental analysis", as it really is, an initial test of a monitoring system. Also because of this last issue, we consider it as an initial step to continue working on this journal, since the next objective and future lines of research in this area is to continue monitoring the territory, extending it, in order to continue evaluating this methodology.

On the other hand, changes have been made in section 3.2.3 to improve the technical quality of the article. Thanks to your comment we have seen that two processes were missing to be explained and related to figure 3, which have now been solved.

Thank you very much for your work.